# Probiotics Regulating Inflammation via NLRP3 Inflammasome Modulation: A Potential Therapeutic Approach for COVID-19

**DOI:** 10.3390/microorganisms9112376

**Published:** 2021-11-17

**Authors:** Arezina N. Kasti, Kalliopi D. Synodinou, Ioannis A. Pyrousis, Maroulla D. Nikolaki, Konstantinos D. Triantafyllou

**Affiliations:** 1Department of Nutrition and Dietetics, Attikon University General Hospital, 12462 Athens, Greece; kastiare@med.uoa.gr (A.N.K.); kall.synodinou@gmail.com (K.D.S.); ipyrousis@gmail.com (I.A.P.); maroullanikolaki@gmail.com (M.D.N.); 2Medical School, University of Patras, 26504 Patras, Greece; 3Hepatogastroenterology Unit, 2nd Department of Propaedeutic Internal Medicine, Medical School, National and Kapodistrian University of Athens, Attikon University General Hospital, 12462 Athens, Greece

**Keywords:** probiotics, inflammasomes, NLRP3 activation, mammalians, SARS-CoV-2

## Abstract

Inflammasomes are cytoplasmic multiprotein complexes formed by the host’s immune system as a response to microbial infection and cellular damage. Many studies have revealed various regulators of NOD-, LRR-, and pyrin domain-containing protein 3 (NLRP3) inflammasome activation, while it has been recently shown that NLRP3 is implicated in COVID-19 pathogenesis. At the same time, probiotics counteract the inflammatory process and modulate cytokine release, thus influencing both innate and adaptive immune systems. Herein, we review the immunomodulatory potential of probiotics on the assembly of NLRP3 inflammasome, as well as the pathophysiological mechanisms supporting the use of probiotic bacteria for SARS-CoV-2 infection management, presenting evidence from preclinical studies of the last decade: in vivo, ex vivo, and mixed trials. Data show that probiotics intake is related to NLRP3 inflammasome attenuation and lower levels of inflammation markers, highlighting the beneficial effects of probiotics on inflammatory conditions. Currently, none of the ongoing clinical trials evaluating the effectiveness of probiotics intake in humans with COVID-19 has been completed. However, evidence from preclinical studies indicates that probiotics may block virus invasion and replication through their metabolites, bacteriocins, and their ability to block Angiotensin-Converting Enzyme 2 (ACE2), and by stimulating the immune response through NLRP3 inflammasome regulation. In this review, the beneficial effects of probiotics in the inflammatory process through NLRP3 inflammasome attenuation are presented. Furthermore, probiotics may target SARS-CoV-2 both by blocking virus invasion and replication and by stimulating the immune response through NLRP3 inflammasome regulation. Heterogeneity of the results—due to, among others, different bacterial strains and their metabolites, forms, dosage, and experimental designs—indicates the need for more extensive research.

## 1. Introduction

Viral infection triggers host innate immune responses indicated by cytokine production and inflammasome activation [1]. Inflammasomes are cytoplasmic multiprotein complexes which consist of a sensor protein, the adaptor molecule apoptosis-associated speck-like protein containing a caspase recruitment domain (ASC), and the effector protein caspase-1. Active caspase-1 proceeds to cleave the precursors pro-interleukin (IL)-1β and pro-IL-18 into mature forms IL-1β and IL-18, respectively, inducing pyroptosis. Inflammasomes are vital for the maintenance of intestinal homeostasis and gut-associated physiological inflammatory responses in humans. The NLRP3 (NLR family pyrin domain containing 3) inflammasome is assembled in both gut immune and epithelial cells [2] and it is the most widely studied inflammasome.

Probiotic bacteria (PB) are live microorganisms that, when administered in adequate amounts, confer a health benefit on the host [3,4]. *Lactobacilli* and *Bifidobacteria* are the most common probiotics, but the yeast *Saccharomyces boulardii* and *Bacillus* species are also widely known [4]. PB—generally regarded as safe (GRAS)—have beneficial effects in various clinical conditions, such as the prevention of antibiotic-associated diarrhea, constipation, necrotizing enterocolitis, sepsis, and allergies in infants, while recently have shown promising results in oral health and periodontal therapy [5,6,7,8,9,10,11]. Lactic acid bacteria (LAB) can prevent gastrointestinal dysbiosis and reduce the risk of developing secondary infections, while other strains and their metabolites exhibit antiviral activities [12,13], such as enhancement of the barrier function by stimulating mucin secretion, antimicrobial activity by competing with microbial pathogens for nutrients and adhesion to epithelial cells producing antimicrobial substances such as bacteriocins and stimulation of mucosal epithelial cells to secrete defensins. Moreover, PB have immunomodulatory activity by interacting with dendritic cells (DCs), monocytes, and lymphocytes (Figure 1) [14].

It is well documented that PB can regulate inflammation in two ways: (1) indirectly, by producing short-chain fatty acids (SCFAs) and (2) directly, by binding to innate immune system receptors Toll-like (membrane glycoproteins TLR 2, 4, 9) and by triggering important signaling pathways. One of these pathways includes the transcription factor NF-κB, which is associated with the activation of NOD-like receptors (nucleotide-binding oligomerization domain-like receptors), that affect the formation of inflammasomes, thus the inflammatory response [15]. Overactivation of NLRP3 inflammasome has been linked to the pathogenesis of inflammatory bowel disease, cryopyrin-associated periodic syndromes, type 2 diabetes mellitus, atherosclerosis, and neurodegenerative diseases [16]. PB interact with the host by germline-encoded host sensors, namely, pattern recognition receptors (PRRs) [17], that detect microbial structures—pathogen-associated molecular patterns (PAMPs) and components of the host’s cells released during cell damage or death named damage-associated molecular patterns (DAMPs) [18,19]. PRRs are expressed by most innate immune effector cells (dendritic cells, macrophages, monocytes, neutrophils, and epithelial cells). The mechanism of triggering the immune response depends on the kind of antigenic molecule [20]. Many recent studies focus on the complex mechanisms of NLRP3 activation stemming from pathogenic bacteria and their products, such as extracellular ATP and pore-forming toxins (Figure 2).

In this review, we aim to investigate the impact PB have on stimulating the assembly of NLRP3 inflammasome, as well as the pathophysiological mechanisms supporting the use of PB in COVID-19.

We assessed preclinical studies focusing on the activation of NLRP3 by PB during the decade 2010–2020. A literature research using the terms “probiotics”, “inflammasome” and “NLRP3” and their combination in PubMed and Cochrane databases revealed twenty-two publications in English language. We used evidence from the original articles only and we included studies identified by manual search of the reference lists of the aforementioned articles. Similarly, we assessed the available literature in PubMed and Cochrane databases and in ClinicalTrials.gov (accessed on 3 December 2020) using the terms “NLRP3, COVID-19, probiotics” “SARS-CoV-2”, and their combination on the use of probiotics as immunomodulators of COVID-19 infection management through NLRP3 inflammasome manipulation.

## 2. Probiotic Bacteria and NLRP3 Activation

We did not detect any publication investigating the role of PB on NLRP3 inflammasome activation in humans in vivo. A total of fifteen studies (in vivo, ex vivo) providing data regarding the NLRP3 inflammasome stimulation by PBs in a variety of cellular and animal models were included in this review. More specifically, four studies on mammals (three porcine, one canine), seven on laboratory animals (four rats, three mice), one on rodents (Syrian hamsters), and three on cell cultures, in which inflammatory biomarkers were measured. Treatment included probiotic supplements (regardless of viability, strains, forms, dosages, and duration of treatment). The effect of probiotics on alterations in inflammation markers (IL-1β, TNF-a, IL-6, IL-18) and any immunomodulatory potential related to intervention were considered as endpoints.

Heterogeneity was the main feature of the included studies. Different types of trials, interventions, subjects, and cell cultures and a variety of PB strains and metabolites, dosages, and duration precluded adequate studies classification (Table 1, Appendix A).

Pigs fed on either a basal diet or a *Clostridium butyricum*-supplemented diet were given orally enterotoxigenic *Escherichia coli* (ETEC) K88 or saline. The results of this Chinese study showed that *C. butyricum* decreased IL-1β and IL-18 levels in serum and gut tissue, whereas IL-10 levels were increased. *C. butyricum* promoted the accumulation of intestinal NLRP3 mRNA and inhibited ETEC K88-induced caspase-1 and NLRP3 increase [35]. In an Italian study that investigated behavioral and obesity effects in hamsters fed with high fat diet (HFD), IL-1β, NLRP3, caspase-1, and NF-kB levels were measured in the presence or absence of a multispecies probiotic formulation. Hamsters were subjected to unpredictable chronic mild stress. Consequently, PB decreased hypothalamic expression and hematic circulating levels of all above inflammatory markers, while HFD increased them [30]. In a study published in 2020, three purified bacteriocins produced by *L. helveticus*: PJ4, L. brevis: DT24, and *L. animalis*: TSU4 were administered in mice as a treatment strategy for HFD induced obesity. Subjects were divided into five groups and all groups were fed with HFD—except for controls. Three HFD groups received three bacteriocins, respectively. Results showed that the inflammatory mediators (IL-1β, IL-6, TNF-α) were significantly increased in the HFD group without bacteriocins administration. The DT24 group did not show any change in cytokines levels, while PJ4 and TSU4 groups showed a significant reduction; PJ4 was more effective than TSU4 in decreasing inflammatory biomarkers. Notably, increased NLRP3 expression was observed in all groups, especially in HFD compared to controls [41]. University of Florida researchers evaluated the effects of *Lactobacillus johnsonii* N6.2 in combination with rosmarinic acid, a natural antioxidant with anti-inflammatory properties, on inflammasome assembly in ileal tissue of diabetes-prone rats. Their findings confirm that *L. johnsonii* suppresses NLRP3 and caspase-1 maturation lowering overall intestinal inflammation [31]. When the effect of oral preadministration of *L. johnsonii* L531 in piglets with *Salmonella infantis*-induced enteritis was evaluated, *S. infantis* activated NLRP3 and NF-κB signaling in the jejunum and ileal tissue, *L. johnsonii* L531 administration before the challenge reduced the severity of intestinal inflammation and prevented the excessive expression of NLRP3 and caspase-1 through the elimination of damaged mitochondria and accelerated autophagic degradation [33]. Similarly, in rats with cecal ligation and puncture-induced sepsis, the administration of a different strain of *Lactobacillus* (*L. rhamnosus* GG, LGG) decreased IL-1β, NLRP3, IL-6, and TNF-a levels in liver tissues indicating the anti-inflammatory role of LGG. Moreover, *B. fragilis* ZY-312 probiotic therapy reduced liver injury following experimental sepsis in neonatal rats with necrotizing enterocolitis (NEC) induced by *Cronobacter sakazakii* [4]. Fan et al. evaluated the effects of *C. sakazakii* on intestinal barrier function and the protective role of ZY-312. They showed that the expression of NLRP3, caspase-1 (p10, p20), IL-1, and gasdermin D were significantly increased in the *C. sakazakii* group. Meanwhile, ZY-312 ameliorated the deleterious effects of *C. sakazakii* on intestinal integrity and attenuated clinical symptoms (weight loss, loss of appetite, abdominal flatulence) and intestinal inflammation [40]. The anti-inflammatory effects of *Roseburia intestinalis*-derived flagellin (a subunit protein of the flagellum, a whip-like appendage that allows bacterial motility) have been investigated in a dextran sulfate sodium (DSS) induced colitis model in mice [43]. The mRNA levels of NLRP3 and IL 1β were upregulated in the DSS group compared with the control group, but treatment with probiotic Roseburia intestinalis flagellin remarkably alleviated the intestinal inflammation inhibiting the increase of proinflammatory cytokines (IL 1β, IL 18, IL 6, and TNF α) levels in serum and decreasing the NLRP3 activation in colonic tissues.

Chung et al. [36]. assessed the immunomodulatory effects of heat-killed *Enterococcus faecalis*, a commensal Gram^+^ lactic acid bacterium, and its potential protective role on intestinal inflammation in murine models of DSS-induced colitis and colitis-associated colorectal cancer. *E. faecalis* ameliorated the severity of inflammation and attenuated NLRP3 activation in THP-1-derived macrophages while inducing the expression of pro-and mature IL-1β but did not affect the amount of active caspase-1 [36]. In another detailed analysis, microbe-derived antioxidant (MA) fermented by *Bacillus subtilis*, *Lactobacillus*, and beer yeast was used in mother rats and offspring. MA supplementation attenuated HFD-induced NLRP3 activation in the liver and decreased IL-1β and IL-18 gene expression in HFD-induced hepatic lipid disorders during pregnancy and lactation and improved hepatic function [37].

### 2.1. Ex Vivo Trials

In 2011, investigators cloned and sequenced porcine NLRP3 cDNA isolated from ileal Peyer’s patches. Then, they examined the expression of NLRP3 in diverse tissues (spleen, esophagus, duodenum, jejunum, ileum, Peyer’s patches, colon, and mesenteric lymph nodes (MLNs) from newborn and adult porcine); they also examined the ability of two *Lactobacilli* strains (*L. delbrueckii* subsp. *bulgaricus* NIAI B6 and *L. gasseri* JCM1131T) to evoke the expression of NLRP3 in the gut-associated lymphoid tissues (GALT) of the subjects showing that the two strains of lactic acid bacteria can enhance NLRP3 expression in adult and newborn GALT [38]. Later, one different team investigated the potency of L. *rhamnosus* GR-1 to prevent *E. coli* adhesion and described the effects of *L. rhamnosus* GR-1 on ameliorating *E. coli*-induced mastitis and cell damage in primary bovine mammary epithelial cells (BMECs), as well as the NLRP3 inflammasome activation. They showed that NLRP3 expression and caspase-1 were increased during *E. coli* infection, while *L. rhamnosus* GR-1 pretreatment ameliorated *E. coli*-induced mastitis. Preincubation of BMECs with probiotic strains had no direct killing effect on *E. coli* but reduced the adhesion levels to about 50% of that observed in BMECs infected with *E. coli*. BMECs treated with *L. rhamnosus* GR-1 did not exhibit increased NLRP3 activation compared with untreated controls, whereas IL-6, IL-8, and TNF-a production was downregulated [32]. Treatment with probiotic *Enterococcus faecium* NCIMB 10415 (*E. faecium*) was examined in porcine monocyte-derived dendritic cells (MoDC) infected by ETEC to elicit NLRP3 inflammasome activation [34]. Inflammasome activation normally requires a two-step process (priming and activation) to induce its transcription; in this case, the research team studied the *E. faecium* potency on primed cells with LPS (mono- and co-incubated; *n* = 5 independent experiments) and unprimed cells (mono- and co-incubated; *n* = 4 independent experiments). In a co-incubation experiment, MoDC were pretreated with *E. faecium* for 1 h and then were challenged with ETEC for 1 h as well, as in the ETEC mono-incubation. The expression of NLRP3 components was measured in MoDC after 1.5, 6, and 20 h of stimulation. The complex design of the experiments showed that priming of MoDC with LPS for 3 h induced an increased mRNA expression of IL-1β, IL-18, caspase-1, and NLRP3. In the unprimed cells, the mRNA expression of IL-1β, IL-18, and NLRP3 was significantly increased in cultures incubated with ETEC, but at the 6th and 20th h. The pathogenic ETEC strain stimulates a time-dependent inflammasome response, possibly because LPS is present in the outer membrane of ETEC, unlike the *E. faecium* that did not stimulate NLRP3.

### 2.2. Mixed Trials

In 2015, Schmitz et al. [29]. assessed the intestinal expression of caspase-1, IL-1β, IL-18, and NLRP3 in canines with chronic enteropathy (CE) compared to controls when treated with *E. faecium* in vivo and ex vivo. In in vivo experiments, all groups were fed a standardized diet. Samples were collected from duodenal and colonic biopsies. Results showed that inflammasome-related genes in the duodenum were not significantly different between dogs with probiotic or placebo treatment, whilst they were expressed at a much higher level in the colon than in the duodenum (both in healthy and CE dogs). Interestingly, IL-1β protein expression in CE group was decreased with dietary treatment (not with PB). In ex vivo culture of duodenal biopsies (macrophage DH82 cells), the samples were stimulated with different TLR ligands and *E. faecium*. Incubation with *E. faecium* increased caspase-1 levels compared to stimulation with pure TLR ligands (PBS, Flagellin, Pam3CSK4, LPS), but did not affect the NLRP3 expression independently of disease status. Challenge with pure TLR ligands showed minor effects on mRNA levels of the NLRP3 components apart from IL-18 [29].

In another mixed trial, German scientists investigated the expression of NLRP3 components in sows and piglets’ intestines (jejunal, ileal, and colonic tissues) and analyzed the influence of age and long-term supplementation with the *E. faecium* NCIMB 10,415 [42] on NLRP3 expression. NLRP3 expression in tissues was higher in 29-day-old piglets compared to 70-day-old growing pigs, indicating that age is a factor that affects NLRP3 inflammasome activation. Furthermore, they examined cell cultures (intestinal epithelial cells) which first were challenged with ETEC and then inoculated with *E. faecium* NCIMB 10415. Expression of NLRP3 was slightly higher in epithelia mono-incubated with *E. faecium* or ETEC compared with epithelia incubated with *E. faecium* and ETEC whereas IL-1β and IL-18 did not differ significantly between the treatment groups but tended to be higher in epithelia incubated with ETEC.

To sum up, most of the above-mentioned trials showed NLRP3 attenuation and decreased levels of inflammation markers -caspase-1, IL-1β, IL-6, IL-18, and TNF-a-after PB administration/incubation (Table 1). The age of the subjects, the exact time in which the samples were analyzed, and the condition of treatment or pretreatment of different pathogens and challenges are of concern in interpreting the results. Beneficial effects of PB seem to be affected by many factors, including different bacterial strains and their metabolites, forms (viable or nonviable3. Involvement of SARS-CoV-2 in NLRP3 Activation.

Although COVID-19 pathogenesis remains elusive, emerging evidence indicates the role of NLRP3 inflammasome involvement in its pathogenesis [44,45].

In humans, the main clinical manifestations of severe acute respiratory syndrome coronavirus 2 (SARS-CoV-2) infection are severe acute respiratory failure and macrophage activation syndrome (MAS) [46]. The immune response to SARS-CoV-2 is driven by inflammatory alveolar and monocyte-derived macrophages, which are activated by PAMPs and DAMPs, released by infected pneumocytes [47]. The massive release of cytokines, produced by the triggered innate immune system, leads to NLRP3 hyperactivation through viroporins. Viroporins are a group of viral proteins (protein E, open reading frame 3a (ORF3a), and ORF8a) with ion channel activity that takes part in virus replication and disease pathogenesis [48]. SARS-CoV-2 was confirmed to evoke gut inflammation in intestinal epithelial cells (IECs) and epithelial alveolar cells by invading in the same way. The virus invades the cells of the host through angiotensin-converting enzyme 2 (ACE2) receptors, expressed in both the respiratory and the GI tract. The viral protein S (Spike S glycoprotein) is activated by the transmembrane protease serine 2 (TMPRSS2) leading to the proinflammatory cytokine cascade. The downregulation of ACE2 caused by SAR-CoV-2 leads to alter gut microbiota, increased intestinal permeability, and inflammation directly linked to gastrointestinal symptoms and diarrhea in patients with COVID-19 [49,50,51].

It is already known that there is bidirectional crosstalk between gut and lung, named the gut–lung axis, which is involved in immune homeostasis. The underlying mechanism linking dysbiosis of gut microbiota with several respiratory diseases and dysbiosis of the lung microbiota is not fully understood. A potential mechanism could be the leaky gut and the migration of bacterial products or particles to the lung, stimulating the immune response. Additionally, blood or lymphatic mediated circulation of immune cells or inflammatory cytokines from the GI tract to the lung can induce inflammatory response [21]. The antiviral activity of probiotics against common respiratory viruses (including influenza, rhinovirus, and respiratory syncytial virus) is already confirmed by clinical and experimental studies [22,52]. LAB may offer protection from airway infection indirectly, through interaction with GALT, eliciting an enhancement of respiratory immunity. Furthermore, the protective role of probiotics is associated with the activation of proinflammatory natural killer (NK) cells and macrophages within the airway mucosa [53].

## 3. The Antiviral Activity of Probiotic-Produced Metabolites

PB, through the production of antimicrobial substances, such as bacteriocins (proteinaceous products), inhibit bacterial adherence and invasion capacity in the intestinal epithelium [54]. The inhibitory activity of bacteriocins includes effects against pathogens responsible for hospital-acquired infections and many Gram-negative bacteria. They can bind to the cell surface receptors and reduce the virus-induced cytopathic effects at a preincubation condition. Additionally, bacteriocins inhibit virus replication (*Lactobacillus delbrueckii* subsp. *Bulgaricus* 1043 bacteriocin inhibited the replication of influenza virus) blocking receptor sites on host cells and avoid the accumulation of viral particles [14,47,55], as well. Enterocin CRL35, a bacteriocin produced by *Enterococcus mundtii* CRL35, has proven antiviral effects against strains of *Herpes simplex* viruses (HSV)-1 and HSV-2 by inhibiting late stages of replication in vitro [56]. Furthermore, probiotic *Bacillus amyloliquefaciens* produces another bacteriocin, subtilosin, with antiviral properties against HSV and influenza virus. In this context, bacteriocins may have antiviral action against SARS-CoV-2 [57].

Hydrogen peroxide (H_2_O_2_) is a defense mechanism with natural microbicide action preventing contamination by microorganisms. H_2_O_2_ is a non-proteinaceous substance produced by several bacteria, and probiotics. More specifically, studies showed that *Lactobacillus* sp. in the vaginal cavity produces H_2_O_2_ with toxic effects on viruses like HIV and HSV-2. In vitro tests revealed that the amount of H_2_O_2_ was sufficient to inactivate HIV [13,58] and researchers concluded that H_2_O_2_ protect against a variety of viral pathogens and improve antigenicity and immunogenicity [59].

In this context, other metabolites produced by *Lactobacillus plantarum* showed effectiveness against transmissible gastroenteritis virus infection, a member of the family *Coronaviridae*. However, the evidence on mechanisms of action by which these probiotic-produced metabolites exert their specific effects in certain systems or diseases has not yet been elucidated [59].

### 3.1. Probiotics Targeting Angiotensin-Converting Enzyme

In animal models experimentally infected with SARS-CoV-2, the protein sequence and structure of ACE2 are conserved across mammalians [60]. However, until now, there has been no published study in which NLRP3 activation was investigated following SARS-CoV-2 infection in animals. LAB fractions (mainly belonging to *Lactococcus lactis* and *Lactobacillus helveticus*), in in vitro experiments, showed inhibitory activity towards ACE enzymes by blocking the active sites [24,25]. Moreover, the debris of the dead probiotic cells acts as ACE inhibitors too, suggesting that PB could be a potential blocker of ACE and ACE2 receptors which act as a gateway for SARS-CoV-2 to attack GI cells (Figure 3) [22,61]. Recently, another in vitro trial in this field has confirmed that *Lactobacillus plantarum* bacteriocins (Plantaricin W, D, and JLA-9) inhibit the entry of SARS-CoV-2 by blocking ACE2 receptors and virus transcription (targeting on the S protein and blocking RNA polymerase (RdRp)) [21]

### 3.2. Research Response to the Pandemic

The ability of probiotics to inhibit the virus replication is more effective in the early onset of disease [14]. At present, there are eleven ongoing in vivo studies in this field whilst none of them have been completed (Table 2, Home—ClinicalTrials.gov, assessed on 31 December 2020); thus, the available evidence comes from the laboratory.

Research, at the time of the pandemic, poses a lot of challenges for academics. The design of the studies was dramatically rapid and, in some cases, with an impact on quality. The studies below enrolled a small number of participants, whilst heterogeneity was observed in measurements of the outcomes. Another possible drawback is the use of one probiotic strain in some interventions. Multispecies PB seems to be more effective compared to monostrain probiotics. When multispecies PB are combined, they act synergistically and can have a significantly positive effect. Their activity can also be stimulated through symbiosis among different strains [68].

## 4. Conclusions

To conclude, PB may target SARS-CoV-2 in two different ways: by blocking virus invasion and replication through their metabolites, bacteriocins, and their ability to block ACE2 and by stimulating the immune response through NLRP3 regulation. While it is difficult to extrapolate the results from experimental studies with different strains and different types of samples in humans, in most of the above trials, PB were able to shield the host’s immunity system. PB administration as a preventive method and therapeutic modality through the NLRP3 inflammasome manipulation should be tested as a safe and affordable solution.

## Figures and Tables

**Figure 1 microorganisms-09-02376-f001:**
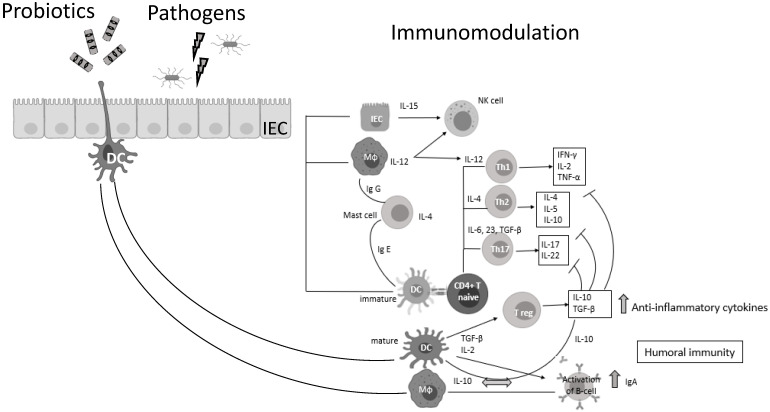
Probiotic bacteria can modulate cytokine release, thus influencing both innate and adaptive immune response. Part of this figure was created with BioRender (https://biorender.com, accessed on 2 October 2021).

**Figure 2 microorganisms-09-02376-f002:**
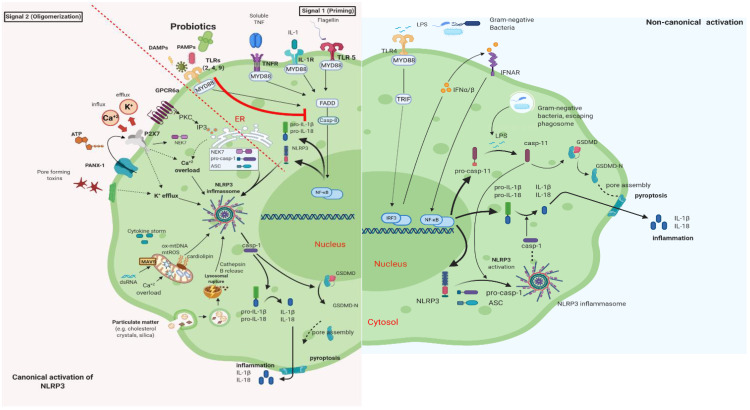
Canonical and Non-Canonical activation of NLRP3 inflammasome. The macrophage in an animal model (transgenic mice-expressing the human angiotensin I-converting enzyme 2 (ACE2) receptor driven by the cytokeratin-18 (K18) gene promoter (K18-hACE2)) [21,22]. (i) The **canonical** pathway of NLRP3 requires a two-step process: priming and activation. **Priming** (signal 1) is the upregulation of the NLRP3 inflammasome components, NLRP3, pro-ILβ, and pro-caspase. This transcriptional upregulation is stimulated by PAMPs (bacterial, fungal, viral) or DAMPs which are recognized by PRRs such as TLRs. Additionally, cytokines such as TNF or IL-1 stimulate this upregulation by engaging TNFR and IL1-R, respectively. The **activation** (signal 2) is provided by various stress signals such as ion or plasma perturbations, ATP, lysosomal rupture, particulate matter, mitochondrial antiviral-signaling protein, mitochondrial reactive oxygen species, ox-mtDNA, etc. Oligomerization of NLRP3 inflammasome activates caspase-1, which in turn cleaves pro-ILβ, pro-IL18, and GSDMD and induces pyroptosis and inflammation [23]. (ii) Gram-negative bacteria (e.g., lipopolysaccharide (LPS) or outer membrane vesicle from bacteria) can activate the **non-canonical** inflammasome pathway that involves NLRP3-dependent caspase-4/5 in humans (known as caspase-11 in murine models) [24]. Caspase-11, when it is secreted at normal rates, protects against bacterial infections, but its excessive activation can cause tissue damage and pyroptosis which is activated in both pathways through cytosolic gasdermin (GSDMD), a member of the gasdermin family [25]. GSDMD cleavage generates a N-terminal domain that is capable of forming plasma membrane pores and a C-terminal domain that acts as an inhibitor of cytolysis [26]. The cleaved N-terminal domain of GSDMD oligomerizes and forms pores on the host cell membrane, leading to pyroptosis and further activation of NLRP3 by triggering K^+^ efflux [27,28]. This figure was created with BioRender (https://biorender.com, accessed on 2 October 2021).

**Figure 3 microorganisms-09-02376-f003:**
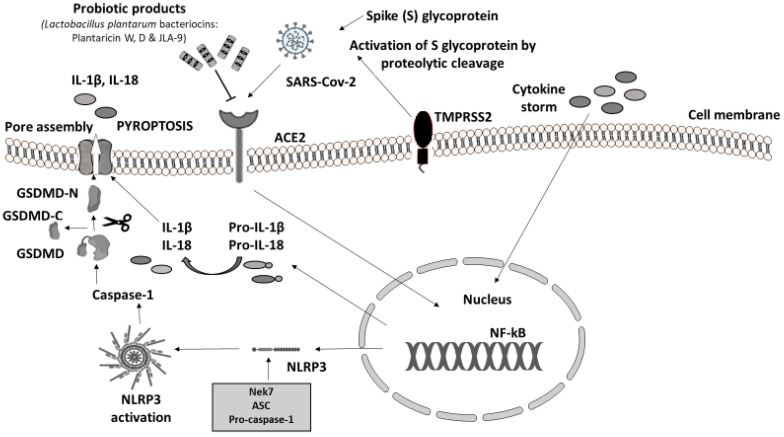
Probiotic bacteria products blocking ACE2 receptors may attenuate COVID-19 infection. Immune response to SARS-CoV-2. Virus invasion is carried out through ACE2 receptor and viral protein S (Spike S glycoprotein) is activated by TMPRSS2 leading to the proinflammatory cytokine cascade. The massive release of cytokines (IL-1β, IL-2, IL-7, IL-8, IFNγ, TNFα, IP10, etc.) from innate immune cells induces the activation of NF-kΒ signaling pathway that regulates the transcription of NLRP3 inflammasome-related components as well as pro-IL-1β and pro-IL-18. As soon as it is formed, NLRP3 inflammasome cleaves pro-caspase-1 and converts it into its active form, caspase-1 [23,62,63]. Subsequently, caspase-1 proceeds with the cleavage of inactive precursors pro-IL-1β and pro-IL-18 to their mature forms IL-1β and IL-18 respectively; at the same time GSDMD is cleaved by the same protease in two domains, an N-terminal and a C-terminal. Following the N-terminal fragment of GSDMD assembles pores in the plasma membrane, inducing pyroptosis and cell death [23]. Experimental studies have shown that SARS-CoV-2 cell entry could be potentially prevented through the antagonistic effect of specific probiotic strains. PB, except for their ability to inhibit the adhesion of pathogens in the intestinal epithelium, compete with pathogens for nutrients and produce antimicrobial substances (e.g., bacteriocins), providing immunomodulatory action and enhancing epithelial barrier function [64]. Their anti-inflammatory effect suppresses cytokine production. Additionally, SARS-CoV-2 evokes inflammation by infecting host cells through the ACE2 receptor and TMPRSS2 inducing proinflammatory cytokine release. *Lactobacillus plantarum* metabolic product (Plantaricin) blocks the entry of SARS-CoV-2 by binding with ACE2 [21,23,65,66,67]. Parts of this figure were created with BioRender (https://biorender.com, accessed on 2 October 2021) and MOTIFOLIO Download Free Sample (https://www.motifolio.com/sampleslides.html, accessed on 2 October 2021).

**Table 1 microorganisms-09-02376-t001:** Trials (in vivo, ex vivo) with probiotic administration and NLRP3 activation in animals and cell cultures.

Author, Year of Publication	Subjects	Treatment	NLRP3	Mechanism of the Antiviral Activity
Schmitz S. et al., 2015 (in vivo)	Canine	1 × 10^7^ CFU *Enterococcus faecium*	Suppression	Reduction of the expression of mRNA transcripts for NLRP3
(ex vivo) [29]				No significant difference in mRNA expression of NLRP3 and IL-1β
Avolio et al., 2018 (in vivo) [30]	Syrian Hamsters	Mix of probiotics, 3 g of which contained: 1.5 × 10^10^ CFU *Streptococcus thermophilus* 1.5 × 10^10^ CFU *Lactobacillus bulgaricus* 1.5 × 10^10^ CFU *Lactococcus lactis subsp. lactis* 1.5 × 10^10^ CFU *Lactobacillus acidophilus* 1.5 × 10^10^ CFU *Streptococcus thermophiles* 1.5 × 10^10^ CFU *Lactobacillus plantarum* 1.5 × 10^10^ CFU *Bifidobacterium lactis* 1.5 × 10^10^ CFU *Lactobacillus reuteri*	Suppression	Decrease in expression levels of NLRP3 in hypothalamus, along with the circulating levels in blood
Teixeira et al., 2018 (in vivo) [31]	Rats	1 × 10^8^ CFU *L. johnsonii* N6.2	Suppression	Reduction in caspase—1 expression and maturation
Ding et al., 2019 (in vivo) [4]	Rats	1 × 10^9^ CFU/ml *Lactobacillus rhamnosus GG*	Suppression	Reduction in mRNA and protein levels of pro-inflammatory cytokines IL-1β, NLRP3, TNF-a, and IL-6 in the liver
Wu et al., 2016 (ex vivo) [32]	Bovine	3 × 10^7^ CFU *Lactobacillus rhamnosus*	Suppression	Decrease of the production of IL-8, IL-6, and TNF-a associated with synergistic responses of TLR-, NLR-, and NLRP3-mediated signaling
Xia et al., 2020 (in vivo) [33]	Porcine	1 × 10^9^ CFU/mL, 10 mL/day *Lactobacillus johnsonii L531*	Suppression	Suppression of inflammasome activation through the elimination of damaged mitochondria and accelerated autophagic degradation
Loss et al., 2018 (ex vivo) [34]	Porcine	*Enterococcus faecium*	Non activation	*E. faecium* does not influence or only minimally affects the NLRP3 inflammasome pathway
Li et al., 2018 (in vivo) [35]	Porcine	3 × 10^8^ CFU *Clostridium butyricum*	Non activation	Inhibition of the ASC-independent NLRP3 inflammasome signaling pathway
Chung et al., 2019 (in vivo) [36]	Mice	17 mg/kg *Enterococcus faecalis KH2*	Suppression	Attenuation of bacteria-Induced NLRP3 activation by decreasing bacterial phagocytosis
Luo et al., 2019 (in vivo) [37]	Rats	microbe-derived antioxidant (MA) (fermented by *Bacillus subtilis, Lactobacillus*, Beer yeast)	Suppression	MA decreased gene expression of NLRP3, IL-1β, and IL-18
Tohno et al., 2011 (ex vivo) [38]	Porcine	*Lactobacillus delbrueckii* subsp. *bulgaricus* and *Lactobacillus gasseri*	Appropriate activation	*Lactobacillus* strains promote the NLRP3 expression via TLR and NOD-mediated signaling, inducing of appropriate NLRP3 activation
Bai et al., 2020 (in vivo) [39]	Mice	Bacteriocins produced by *L. helveticus*: PJ4, *L. brevis:* DT24 and *L. animalis*: TSU4	Suppresion	Reduction in TNF-a, IL-1β, IL-6 because of PJ4 supplementation
Fan et al., 2019 (in vivo) [40]	Rats	*Bacteroides fragilis* ZY-312	Suppresion	Reduction in NLRP3, caspase-1, IL-1β, and GSDMD
Wu et al., 2020 (in vivo) [41]	Mice	*Roseburia intestinalis*	Suppresion	Reduction in NLRP3 activation, caspase -1, GSDMD, and pyroptosis in THP-1 macrophages.
Kern et al., 2017 (in vivo)	Porcine	*Enterococcus faecium*	No difference	No difference in mRNA expression of NLRP3, caspase-1, IL-1β, and IL-18 in the duodenum and colon
(ex vivo) [42])				No significant difference in NLRP3 expression

CFUs: colony forming units; GSDMD: gasdermin; ASC: apoptosis-associated speck-like protein; TLR: Toll-like receptor; TNF-a: tumor necrosis factor alpha.2.1. In Vivo Trials.

**Table 2 microorganisms-09-02376-t002:** Ongoing in vivo studies evaluating probiotic treatment in patients with COVID-19 infection (ClinicalTrials.gov accessed on 31 December 2020).

Number of Clinical Trial	Study Title	Sample Size	Condition/Disease	Group	Interventions	Duration	Measurements	Outcomes
04621071	Efficacy of probiotics in reducing duration and symptoms of COVID-19	84	COVID-19	**Control:** Placebo **Intervention:** 2 probiotic strains	Placebo: (potato starch and magnesium stearate) Probiotic: (2 strains 10 × 10^9^ CFU	25 days	Questionnaires (socio-demographic, medical (weight, height, general health, current medication, symptoms), food intake etc.). Optional collection of saliva and stool samples.	Duration of symptoms. Number of days before symptoms disappear.
04458519	Efficacy of intranasal probiotic treatment to reduce severity of symptoms in COVID-19 infection	40	COVID-19	**Control:** Placebo **Intervention:** Probiotics	Placebo: Saline solution Probiotic: *Lactococcus lactis* W136 2.4 billion CFU given intranasally	4 weeks	Visual Analogue Scale (VAS)	Change in severity of COVID-19 infection. Number of days with any symptom of COVID-19 infection ≥ to 35 as measured on VAS at the 28-day endpoint.
04390477	Study to evaluate the effect of a probiotic in COVID-19	40	COVID-19	**Control:** No treatment **Intervention:** Probiotics	Probiotic pill 1 × 10^9^ CFU	30 days	Patients’ status	Subjects discharged from ICU.
4366180	Evaluation of the probiotic *Lactobacillus Coryniformis* K8 on COVID-19 prevention in healthcare workers	314	COVID-19	**Control:** Placebo **Intervention:** Probiotics	Placebo: maltodextrin Probiotic: *Lactobacillus* K8 per day (3 × 10^9^ CFU/day)	2 months	SARS-CoV-2 detection by PCR or antigen test	Incidence of SARS-CoV-2 infection in healthcare workers
04666116	Changes in viral load in COVID-19 after probiotics	96	COVID-19	**Control:** No dietary administration (probiotics). Medication agreed by the hospital committee **Intervention:** Medication agreed by the hospital committee and nutritional supplement (probiotics)	Dietary supplementation with strains from *Bifidobacterium longum*, *Bifidobacterium animalis* subsp. *Lactis* and *Lactobacillus rhamnosus*	1 year	SARS-CoV-2 detection by PCR	Viral load during the period of admission
04366089	Oxygen-Ozone as adjuvant treatment in early control of COVID-19 progression and modulation of the gut microbial flora	152	COVID-19 SARS-CoV-2 Pneumonia, Viral Coronavirus Infection	**Control:** Drugs (Standard of care) **Intervention:** Oxygen-ozone, Probiotics and Drugs (Standard of care)	Drugs: Azithromycin, Hydroxychloroquine Oxygen-ozone therapy + Probiotics *Streptococcus thermophilus DSM322245, Bifidobacterium lactis DSM 32246, Bifidobacterium lactis DSM 32247, Lactobacillus acidophilus DSM 32241, Lactobacillus helveticus DSM 32242, Lactobacillus paracasei DSM 32243, Lactobacillus plantarum DSM 32244, Lactobacillus brevis DSM 27961* (200 billion) + Drugs (Azithromycin, Hydroxychloroquine)	21 days	Delta neutrophil index (Reflects the ratio of circulating immature neutrophils)	Delta index in the number of patients requiring orotracheal intubation despite treatment
04517422	Efficacy of *L. Plantarum and P. Acidilactisi in* adults with SARS-CoV-2 and COVID-19	300	COVID-19	**Control:** Placebo **Intervention:** Probiotics	Placebo: Combination of maltodextrin (E1400, qs) in a vegetable hydroxymethylpropyl-cellulose capsule Probiotic: Combination of *Lactobacillus plantarum* CECT7481, *Lactobacillus plantarum* CECT 7484, *Lactobacillus plantarum* CECT 7485, and *Pediococcus acidilactici* CECT 7483	30 days	WHO Clinical Progression Scale	Severity progression of COVID-19
04462627	Reduction of COVID-19 transmission to health care professionals	500	COVID-19	**Intervention 1:** COVID-19 positive patients **Intervention 2:** COVID-19 negative patients **Intervention 3:** Untested healthy volunteers	Probiotic (intervention 3)	3 weeks	Determination of the blood group (ABO/LE)	Anti-A antibody concentration Anti-B antibody concentration Blood group
04420676	Synbiotic therapy of gastrointestinal symptoms during COVID-19 infection	108	COVID-19	**Control:** Placebo **Intervention:** Probiotics	Placebo: matrix containing maize starch, maltodextrin, inulin, potassium chloride, hydrolysed rice protein, magnesium sulphate, fructooligosaccharide (FOS), enzymes (amylases), vanilla flavour and manganese sulphate. Probiotic: *Bifidobacterium bifidum W23, Bifidobacterium lactis W51, Enterococcus faecium W54, Lactobacillus acidophilus W37, Lactobacillus acidophilus W55, Lactobacillus paracasei W20, Lactobacillus plantarum W1, Lactobacillus plantarum W62, Lactobacillus rhamnosus W71* and *Lactobacillus salivarius W24*	30 days	Feces	Duration of diarrhea (defined as days with 3 or more loose stools)
04399252	Effect of Lactobacillus on the microbiome of household contacts exposed to COVID-19	1000	Microbiome/COVID-19	**Control:** Placebo **Intervention:** Probiotics	*Lactobaciltus rhamnosus GG* Placebo Probiotic: *Lactobacillus rhamnosus GG*	28 days	Fecal samples	Change in Shannon Diversity Index

## Data Availability

Not applicable.

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
