# Peer review of "Probiotics Regulating Inflammation via NLRP3 Inflammasome Modulation: A Potential Therapeutic Approach for COVID-19"

_microorganisms, 2021, doi:10.3390/microorganisms9112376_

Round 1

Reviewer 1 Report

The information is interesting, relevant  and timely.

  1. However, there is a significant remark. Tables are overloaded with information. It would be better to put them in the accompanying materials section and make an additional generalizing literary data table with grouping of probiotics that cause activation or  suppression of the  NLRP3 inflamasome. This would make it possible to choose more promising probiotics suitable for COVID-19 therapy in the future  Of course, it is also necessary to clarify the mechanism of their antiviral activity. In any case, despite the difference in the conditions of the experiments reported by the authors of the review, the separation of probiotics by the analyzed parameters, at least in the form of an enumeration, would be advisable.
  2.   The manuscript contains a sufficiently large number of inaccuracies that should be corrected. They are indicated in the notes after the "editing" of the text.
  3.  It can be noted, also, a wider list of viruses against which bacteriocins produced by probiotic cultures. In addition, some probiotics produce peroxide and other metabolites, which have an antiviral effect. Also, the effect under consideration is not the only one. Moreover, each probiotic has a a specific set of tools to influence on immunity, including antiviral action due to adaptive and innate immunity.

Author Response

Reviewer 1

  1. However, there is a significant remark. Tables are overloaded with information. It would be better to put them in the accompanying materials section and make an additional generalizing literary data table with grouping of probiotics that cause activation or suppression of the NLRP3 inflammasome. This would make it possible to choose more promising probiotics suitable for COVID-19 therapy in the future. Of course, it is also necessary to clarify the mechanism of their antiviral activity. In any case, despite the difference in the conditions of the experiments reported by the authors of the review, the separation of probiotics by the analyzed parameters, at least in the form of an enumeration, would be advisable.

Response: We made a new table, more readable, according to your valuable advice. New Table 1 includes in vivo and ex vivo trials with specific probiotic strains to suppress NLRP3 and describes, in short, the mechanism of their action. You can see the new version at the end of chapter 2.3. 

The previous overloaded table was added as supplemental material at the end of this paper. Even if the specific effects of probiotics in certain systems have not been elucidated yet, we tried to collect their possible involvement in NLRP3 activation. 

  1. The manuscript contains a sufficiently large number of inaccuracies that should be corrected. They are indicated in the notes after the "editing" of the text.

Response:  We made all the suggested changes that concern comments about missteps in the main text like uppercase, abbreviations, etc.  

  1. It can be noted, also, a wider list of viruses against which bacteriocins produced by probiotic cultures. In addition, some probiotics produce peroxide and other metabolites, which have an antiviral effect. Also, the effect under consideration is not the only one. Moreover, each probiotic has a specific set of tools to influence on immunity, including antiviral action due to adaptive and innate immunity.

Response: We added a new paragraph in chapter 3.1: “The antiviral activity of probiotic- produced metabolites”. Thanks to your advice, we included a list of bacteriocins with antiviral action, and we mentioned the antiviral effect of hydrogen peroxide too.

Reviewer 2 Report

Manuscript of Kasti et al. reviews the main literature on the topic and present the potential beneficial effects of probiotics in contrasting SARS-COV2 virus invasion and in dampening the infection thorough the NLRP3 inflammasome acitivty.

Manuscript is well presented and the topic is of interest, however some minor changes should be done to improve the quality of the review:

  1. Although studies in table 2 are of primary interest, they are not cited in the text. Data emerging from literature should be adequately presented and discussed in the main text. The structure of the manuscript should be revised: the first part presents the effects of probiotics on NLRP3 inflammasome, then Authors discuss about the involvement of SARS-CoV2 in NLRP3 activation, and the last section presents the potential mechanisms of probiotics in contrasting infection. To sustain the main hypothesis of the use of probiotics for COVID-19, clinical studies (table 2) should be presented at the end and data emerging from literature should be discussed and commented.  
  2. Usually "trials" refer to clinical trials, performed in human, to evalutate the efficacy and safety of a new or old compund for a different indication. It should be more appropriated referring to "in vivo/in vitro/ex vivo studies".
  3. In figure 2, Authors represent  the canonical and non canonical activation of NLRP3 but also the anti-inflammatory effects of Lactobacillus plantarum in contrasting SARS-CoV2 (as already presented in figure 3). To simplify the figure 2, the part of PB-virus should be moved in figure 3. 

Author Response

  1. Although studies in table 2 are of primary interest, they are not cited in the text. Data emerging from literature should be adequately presented and discussed in the main text. The structure of the manuscript should be revised: the first part presents the effects of probiotics on NLRP3 inflammasome, then Authors discuss about the involvement of SARS-CoV2 in NLRP3 activation, and the last section presents the potential mechanisms of probiotics in contrasting infection. To sustain the main hypothesis of the use of probiotics for COVID-19, clinical studies (table 2) should be presented at the end and data emerging from literature should be discussed and commented.

Response: Thank you for the remark; We made the necessary changes to fulfil your request and added a new chapter with the title: “research response to the pandemic”

  1. Usually "trials" refer to clinical trials, performed in human, to evaluate the efficacy and safety of a new or old compound for a different indication. It should be more appropriated referring to "in vivo/in vitro/ex vivo studies"

Response: It was corrected

  1. In figure 2, Authors represent the canonical and non-canonical activation of NLRP3 but also the anti-inflammatory effects of Lactobacillus plantarum in contrasting SARS-CoV2 (as already presented in figure 3). To simplify the figure 2, the part of PB-virus should be moved in figure 3. 

Response: Thank you for the suggestion; Done as requested.

Reviewer 3 Report

Dear Authors, 

thank you very much for your work dealing with the possible use of probiotics for the treatment of COVID-19. Obviously, this is a very interesting and actual topic due to the pandemic, with a significant importance from a clinical point of view.

I just have a piece of advice to improve the quality of yourk work. I suggest to add in the introduction a bief discussion about the actual uses of probiotics in medicine and dentistry. In particular, one of the most recent application of probiotics is for the treatment of periodontal disease. You could for example refer to the following article:

Butera A, Gallo S, Maiorani C, Molino D, Chiesa A, Preda C, Esposito F, Scribante A. Probiotic Alternative to Chlorhexidine in Periodontal Therapy: Evaluation of Clinical and Microbiological Parameters. Microorganisms. 2020 Dec 29;9(1):69. doi: 10.3390/microorganisms9010069. PMID: 33383903; PMCID: PMC7824624.

After this minor revision, I suggest this article to be published in the journal.

Congratulations to the Authors again!

The reviewer 

Author Response

  1. I just have a piece of advice to improve the quality of your work. I suggest to add in the introduction a brief discussion about the actual uses of probiotics in medicine and dentistry. In particular, one of the most recent application of probiotics is for the treatment of periodontal disease.

Response: Thank you for the suggestion; Done as requested.